# Graphene-Based Magnetic Nanoparticles for Theranostics: An Overview for Their Potential in Clinical Application

**DOI:** 10.3390/nano11051073

**Published:** 2021-04-22

**Authors:** Teresa Lage, Raquel O. Rodrigues, Susana Catarino, Juan Gallo, Manuel Bañobre-López, Graça Minas

**Affiliations:** 1Advanced (Magnetic) Theranostic Nanostructures Lab, Health Cluster, INL—International Iberian Nanotechnology Laboratory, Avenida Mestre José Veiga, 4715-330 Braga, Portugal; pg33051@alunos.uminho.pt (T.L.); juan.gallo@inl.int (J.G.); manuel.banobre@inl.int (M.B.-L.); 2Center for MicroElectromechanical Systems (CMEMS-UMinho), Campus de Azurém, University of Minho, 4800-058 Guimarães, Portugal; scatarino@dei.uminho.pt

**Keywords:** graphene, magnetic nanoparticles, graphene-based nanomaterials, theranostic

## Abstract

The combination of diagnostics and therapy (theranostic) is one of the most complex, yet promising strategies envisioned for nanoengineered multifunctional systems in nanomedicine. From the various multimodal nanosystems proposed, a number of works have established the potential of Graphene-based Magnetic Nanoparticles (GbMNPs) as theranostic platforms. This magnetic nanosystem combines the excellent magnetic performance of magnetic nanoparticles with the unique properties of graphene-based materials, such as large surface area for functionalization, high charge carrier mobility and high chemical and thermal stability. This hybrid nanosystems aims toward a synergistic theranostic effect. Here, we focus on the most recent developments in GbMNPs for theranostic applications. Particular attention is given to the synergistic effect of these composites, as well as to the limitations and possible future directions towards a potential clinical application.

## 1. Introduction

At the nanoscale (1–100 nm) materials exhibit unique size dependent properties which are not achievable in bulk materials [1,2,3]. In a simplistic way, a bulk ferromagnetic material can be observed as several small regions (domains) that are spontaneously magnetized. However, when the particle size of a ferro- or ferrimagnetic material is reduced at diameters near to the monodomain, these nanoparticles are often in a superparamagnetic state at room temperature, meaning that magnetization decreases to zero when the magnetic field that has been applied is removed [4,5,6]. This ability to interact with external magnetic fields allows them to be remotely manipulated or controlled, thus enabling a plethora of possibilities in the development of biomedical technologies aimed at improving the understanding, diagnosis and treatment of different diseases [7]. Magnetic nanoparticles (MNPs) offer a diverse range of applications, from magnetic resonance imaging (MRI), to drug delivery or magnetic hyperthermia [8]. The first MNPs, approved by the US Food and Drugs Administration (FDA) for clinical application, date from 1996, and consisted in the use of magnetic iron oxide nanoparticles to serve as negative contrast agents in MRI to enhance tumor detection in the liver [9,10]. Nevertheless, MNPs can present some disadvantages, such as self-aggregation, toxicity and low performance for biofunctionalization [11]. Foreseeing the need for improved nanomaterials to suppress these limitations, numerous composite magnetic nanosystems and strategies have been developed in the last decade for nanomedicine. Among them, carbon-based nanomaterials are being explored due to their many interesting physicochemical properties different from those of bulk carbon materials like diamond, graphite, fullerenes and nanotubes [1,2,3]. Since 2004, when Geim and Novoselov discovered graphene (a flat monolayer of carbon atoms, also known as graphene monolayer, tightly packed into a 2D honey lattice), many other graphene derivatives, such as graphene oxide (GO), reduce graphene oxide (rGO), and graphene quantum dots (GQD), have attracted great deal of global attention [12,13,14]. Graphene, defined as a single atomic layer of sp^2^ carbon atoms arranged in a honeycomb lattice, is thus defined as the building block of all other carbon-based materials, their classification mostly dependent on the number of layers and chemical modifications [15].

Overall, carbon atoms have the ability to participate in robust covalent bonds with other carbon atoms and also with other elements in diverse hybridization states (sp, sp^2^, and sp^3^) [16], creating carbonaceous monolayers with unique properties, such as high charge carrier mobility, high chemical and thermal stability and large surface area.

In graphite, i.e., 3D multilayers of graphene where each carbon atom is exclusively attached to other carbon atoms in the same plane forming strong covalent bonds, the carbon multilayers are bound to each other mainly through weak forces, such as Van der Waals. For this reason, graphite is considered as a soft carbon-material, and the previous highlighted properties of graphene-based materials are partially lost.

A particular case of a carbonaceous nanomaterial is the carbon nanotube, which is a tubular arrangement of single or multilayered graphene arranged as a roll, and responsible for their specific properties [17,18].

Graphene-based materials (graphene and graphene derivatives), compared to other carbon-based materials, such as graphite, present a larger surface area, are easier to functionalize and have improved solubility, due to their own unique physicochemical, mechanical, optical, thermal, electronic and biomedical properties [17,19]. The presence of free π electrons and reactive sites for surface reactions offer large room to load and deliver drugs, genes, and proteins towards specific cells, tissues, and organs [18]. Moreover, this material also has specific optical properties, which promote its use in optical absorption in the near infrared (NIR) windows (750−1000 nm for the NIR-I window and 1000–1700 nm for the NIR-II window) [16].

Taking advantage of these characteristics, some studies have demonstrated the potential of combining graphene-based materials and MNPs [11,20]. Thereby, GbMNPs are hybrid combinations of these two materials conjugated in various configurations [21]. This promising nanosystem can collate the advantages and unique properties of each nanomaterial separately, namely high magnetic saturation and superparamagnetism from the magnetic nanoparticles and the high thermal and electronic conductivity, high charge carrier mobility and improved biocompatibility of the graphene-based material [17,19,22]. Additionally, it allows the combination of multiple theranostic functionalities into a single platform.

In this review, we will focus on the latest advances in the use of GbMNPs as theranostic probes. Furthermore, we will discuss the synergistic improvement of these hybrid nanosystems, their current limitations, biocompatibility and cytotoxicity and future directions for their clinical application.

### 1.1. Graphene-Based Magnetic Nanoparticles Configurations

GbMNPs can be categorized into two basic configurations (Figure 1A): (**a**) graphene-based materials encapsulated magnetic nanoparticles (GbEMNPs) and (**b**) graphene-based materials decorated with magnetic nanoparticles (GbDMNPs) [23].

In the first configuration, the core made of magnetic material is covered with a graphene-based shell. In this case, the magnetic nanomaterial can be covered by the carbonaceous shell with spherical or oblong geometry, such as the case of nanotubes, and designed as core–shell or yolk–shell. For biomedical applications, the yolk–shell configuration, i.e., core@void@shell, similar to the one represented in Figure 1A/(a), has attracted much attention due to the hollow space that is created between the magnetic core and the carbonaceous shell. Due to the typical mesoporous characteristic of carbonaceous shells, this interior hollow space can then be used to improve the capability of these nanostructures to load higher drug/gene payloads and serve as super-efficient nanocarriers [11]. Overall, the main advantages of the GbEMNPs’ configuration are the capability to provide protection to the magnetic core from corrosion, prevent potential toxic side-effects caused by the exposure of free magnetic nanoparticles and improve the colloidal stabilization of the magnetic nanoparticles by increasing their hydrophilicity.

In the second configuration, graphene-based materials are decorated with magnetic nanoparticles [23]. Generally, GbMNPs nanosystems are mostly produce in GbDMNPs configurations (Figure 1A/(b)). However, in this case the MNPs are not protected against the environment and vice versa. Thus, biological systems are not protected from potential leaches from these materials, demanding further functionalization.

### 1.2. Preparation and Synthesis of Graphene-Based Magnetic Nanoparticles Hybrids

Over the years, several synthesis protocols have been developed to produce graphene and its derivatives, including mechanical exfoliation, epitaxial growth, and liquid phase exfoliation [24]. Among the most popular techniques used for the synthesis of graphene monolayers are the mechanical exfoliation or chemical vapor deposition (CVD); for GO, the most popular is the oxidation of crystalline graphite followed by dispersion in aqueous medium through sonication or other processes; and for rGO, the most popular is the thermal, chemical or UV treatment of GO under reducing conditions [17]. Likewise, MNPs’ synthesis techniques have also been the focus of intensive research and development, which can be listed in three main classifications, i.e., physical, chemical or biological methods, each one with their own advantages and disadvantages. Among those, chemical methods are the most used, allowing efficient and precise control of particle size, composition and surface chemistry. The most successful and popular ones are the coprecipitation, thermal decomposition and hydrothermal methods [25].

In general, two main strategies can be followed for the synthesis of GbMNPs, namely: the ex situ and in situ methods [26]. In the ex situ method, two main steps are followed. Firstly, the magnetic nanoparticles are synthesized and after that, bonded onto the surface of graphene sheets by covalent or noncovalent modification [27,28,29]. In the in situ method (mostly hydrothermal and solvothermal growth and reduction method), nanocrystallites are produced in the presence of functionalized graphene nanosheets, growing directly onto the graphene surface [27].

GbEMNPs’ heterogenic structures are, in general, generated following ex situ methods using direct routes, such as CVD, or by indirect routes, mostly wet chemical techniques [23]. As an example of this last technique, some of the authors of this revision, Rodrigues et al. [30] presented a protocol for the synthesis of tailor-made yolk-shell graphene-based magnetic nanoparticles envisioning biomedical applications. In this work, the authors used a two-step procedure, where first the magnetic core was synthesized, followed by the formation of the carbonaceous nanoshell. For this second step, the authors used a one-pot strategy of hydrolysis and polymerization of the precursors resorcinol, formaldehyde and TEOS. By increasing the amount of those precursors and maintaining a fixed mass of the magnetic core, the authors obtained different yolk-shell GbMNPs architectures, with different sizes of the hollow cavity and thickness of the carbon-shell, ranging from graphene-based to graphite nanostructured shells [30]. The same authors later published the optimization of GbMNPs for combined hyperthermia and dual thermal/pH stimuli-responsive drug delivery [11] showing the potentiality of this procedure to develop multifunctional nanosystems for medicine.

On the other hand, GbDMNPs can be easily obtained by in situ methods. Most of these nanocomposites are prepared by the reduction of the metal precursors on the surface of graphene-based sheets with the help of reductants, such as NaOH, NaBH_4_ and amines, among others [21]. In a recent study published by Isiklan et al., 2021 [31], a synthesis protocol for the development of gelatin-decorated magnetic graphene oxide nanoplatfom is described for photothermal therapy and drug delivery to achieve theranostics. In this work, magnetic graphene oxide nanosheets were synthesized under inert atmosphere using Fe(acac)_3_ as metal precursor and TREG as reducing agent at high temperature (278 °C). Then, the magnetic graphene oxide was decorated with gelatin following a facile mixing approach, which improved the biocompatibility, colloidal and thermal stability with enhancement of the photothermal conversion efficiency compared to the uncoated magnetic graphene oxide. A more complete and comprehensive overview of the synthesis methods for graphene-based and graphene-based magnetic nanoparticles can be found elsewhere [19,32,33].

Overall, and independent on the selected method used to obtain GbMNPs, it is important that the final nanoproduct can be presented in aqueous colloidal dispersions, ideally monodisperse in size and shape, in order to guarantee reproducibility and high-quality performance [4]. Additionally, the cytotoxicity and biocompatibility to tissues and cells is of the outmost importance, which will be focus of discussion in Section 2.3 of this review.

## 2. Graphene-Based Magnetic Nanoparticles for Theranostics

Theranostics involves the combination of diagnostics and therapeutics in a single platform. Among diagnostic strategies the most relevant are: MRI; positron emission tomography, PET; single-photon emission computed tomography, SPECT; computed tomography, CT; and photoacoustic imaging, PAI. Regarding the therapeutic strategies, the most relevant are drug and gene therapy; photothermal therapy, PTT; magnetic hyperthermia, MHT; and photodynamic therapy, PDT [34]. The combination of these strategies in a single platform results from the integration of various functional components.

From the mid-1990s, nanotechnology has been used in therapeutics to target cancer and since then, nanoparticle-based platforms have gradually increased their complexity and have approached a wider range of diseases and conditions [35]. Nonetheless, due to their increased complexity, theranostic platforms are to date unavailable for clinical use [35,36]. Despite the great performance showed by many of the developed theranostic nanostructures at a preclinical level, only a few nanomaterials have been evaluated in clinical trials [37].

Among the wide variety of nanostructures developed for theranostic purposes, GbMNPs represent a research trend that has attracted increasing attention in the last few years (Figure 2a). However, if the search is narrowed to the keywords (“graphene + magnetic + theranostic” (title/abstract/keywords)) (Figure 2b) we verify that the growth trend continues to exist, but in a more modest and oscillating way.

GbMNPs are able to perform or improve several functions key in the different biomedical applications, namely: (i) signal generation, either through GbMNPs’ intrinsic properties, for example, via interactions with electromagnetic fields in MRI or radiation in fluorescent imaging, or via functionalization with external reporters (e.g., radioactive nuclides for PET or SPECT); (ii) the conversion of electromagnetic energy into thermal or chemical energy for remote manipulation of their surroundings, representative of applications such as PTT, PDT and MHT; (iii) drug delivery and (iv) increased specificity via surface modification with targeting biomolecules or magnetic guidance [38], as exemplified in Figure 1B.

In this review we discuss studies proposing the development of GbMNPs for theranostic applications. These studies are only included if they meet the following criteria: (i) have been published within the last six years and, (ii) present at least one strategy for diagnosis plus one strategy for therapy. The literature search was conducted in electronic databases (i.e., PubMed and Scopus). Table 1 presents a summary of characteristics of GbMNPs proposed for theranostic applications. 

The theranostic modality most often described in the studies highlighted in Table 1 is the combination of MRI with chemotherapy. However, many other combinations are possible and, above all, there is currently a great demand, especially in cancer, for synergic theranostic systems [20]. The following sections will cover representative examples of GbMNPs-based theranostic agents involving several diagnostic and therapeutic modalities. Limitations regarding the use of GbMNPs for clinical applications will also be addressed.

### 2.1. Graphene-Based Magnetic Nanoparticles in Diagnostic

As mentioned above, the diagnostic technique most often described in the studies reviewed is MRI. MRI contrast agents have demonstrated their ability to maximize the difference in relaxation time between healthy and diseased tissues and are classified into *T*_1_ or *T*_2_ contrast agents depending on their magnetic properties and imaging effects [49]. On the one hand, the *T*_1_ performance is mainly associated with a paramagnetic character of the contrast agent coming from transition and lanthanide metal ions within their chemical structure presenting a large number of unpaired electrons [50,51]. On the other hand, a *T*_2_ behavior involves a long-range magnetic interaction with water molecules that it is caused by the magnetic field that a superparamagnetic core is able to induce under an applied magnetic field [52]. In GbMNPs hybrids, the observed shortening in the transverse relaxation time (*T*_2_) [4] comes from the MNPs counterpart, since the graphene component does not exhibit intrinsic paramagnetism or superparamagnetism [53,54,55]. Some GbMNPs combinations [39] show transverse relaxivity (*r*_2_) values superior to those of commercial iron oxide contrast agents Feridex^®^ (*r*_2_ = 130 m M^−1^ s^−1^, at 3T, 37 °C) [56] and Resovist^®^ (*r*_2_ = 200 m M^−1^ s^−1^, at 3T, 37 °C) [57], both superparamagnetic iron oxide NPs approved by FDA. This increase in the *r*_2_ of GbMNPs can be also tuned by, for example, increasing iron oxide nanoparticle loadings, controlling MNPs aggregation or increasing average nanoparticle sizes [57], translating into improved *T*_2_-weighted contrast enhancements.

In a study by Qian et al. the MRI performance of RGO-MnFe_2_O_4_-PEG nanocomposites showed both *T*_1_ and *T*_2_ weighted contrast enhancement [39]. The *r*_1_ and *r*_2_ values were calculated to be 11.74 and 295.48 m M^−1^ s^−1^, respectively. Mice bearing 4*T*_1_ tumor models were IV injected with RGO-MnFe_2_O_4_-PEG nanocomposites and imaged in an MRI scanner at 3.0 T (Figure 3a). *T*_2_-weighted MR images showed a dark contrast effect in the tumor sites, whereas *T*_1_ weighted MRI exhibited brighter contrast in the same areas.

Other imaging modalities can also be integrated in GbMNP, including PET and SPECT, which make use of radioactive tracers and allow for higher sensitivity (10^−10^–10^−11^ M) [39]. In this field, graphene-based nanomaterials play an important role due to their optical properties. In contrast to graphene, GO exhibits robust and reproducible fluorescence, with a range of lateral dimensions, and tunable wavelengths [58]. GbMNPs have been used as MRI and SPECT multimodal imaging probes, providing precise information about the location and size of tumors [39]. In the study by Qian et al., mice bearing 4T1 tumors were also i.v. injected with ^125^I-RGO-MnFe_2_O_4_-PEG nanocomposites (10 mg/mL of RGO-MnFe_2_O_4_-PEG, 200 µCi of ^125^I) and then imaged using a small animal SPECT imaging system. The acquired images were over imposed to those of MRI, showing a higher accumulation of the ^125^I-RGO-MnFe_2_O_4_-PEG nanocomposites at the tumor site (Figure 3).

CT it is also used in the studies reviewed, this strategy is based on differential levels of X-ray attenuation by tissues within the body to produce three dimensional high-contrast anatomic images enabling delineation between various structures [44,45]. In a study of Bi et al., a GO/ZnFe_2_O_4_/UCNPs nanocomposite is presented, abbreviated as GZUC [44]. In vitro CT images of GZUC-PEG at different concentrations are shown in Figure 4a, where the signal intensity is stronger with the increasing concentration of GZUC-PEG. CT imaging in vivo is given in Figure 4c. The images revealed that the CT value of the mouse injected with GZUC-PEG is 247 HU, which is higher than the noninjected mouse (i.e., control) [44]. Photoacoustic tomography (PAT) combines the advantages of optics and ultrasound through exciting the sample with a laser and then detecting the ultrasound wave generated (due to laser absorption and subsequent thermoelastic expansion) [44,59], and was also applied as a bioimaging tool in the same study [44]. The study revealed that GZUC-PEG can reach the tumor region, showing its potential to be used as imaging-guided cancer treatment application using both techniques. Another dual bioimaging strategy was proposed by Gonzales-Rodriguez et al., which proposed a multifunctional graphene oxide/iron oxide nanoparticles for dual fluorescence/MRI imaging and the optical detection of cancerous tissues [47]. Using this strategy, the authors have shown the potential of GO-Fe_3_O_4_ as negative MRI contrast enhancement agent in in vivo studies, and the intrinsic green fluorescence of GO-Fe_3_O_4_ complex to track the efficiency of cell’s internalization in in vitro studies, using the ratios of emission intensity in green (535 nm) to red (635 nm) that enabled them to differentiate cancer cells (MCF-7 and HeLa) from healthy cells (HEK-293), with 4 to 5-fold orders of difference.

In the last few years GbMNPs have assumed a substantial importance in the development of other diagnostic applications, such as biosensors, where GbMNPs have been used, for instance, to enhance the detection efficiency and to monitor biomarkers released from diseased tissues. However, biosensors are, in general, developed with the particular aim to detect analytes and not for the combination with other therapeutic approaches. Therefore, this application was not considered in this review, but more information on this topic can be found through this reference [21].

### 2.2. Graphene-Based Magnetic Nanoparticles in Therapy

The therapeutic modality PTT is based on nanoparticles with a photoabsorbing capability to generate heat under NIR irradiation, thus, inducing hyperthermia and leading to apoptosis on unhealthy cells or tissues while protecting healthy ones [60,61]. Graphene-based structures are potential PTT agents due to their high absorbance in the NIR range. These nanocomposites have the potential to absorb NIR in the spectral region, where water and hemoglobin show low optical absorption (750–1700 nm), which enables maximum light penetration depth without affecting tissues or blood cells [62,63]. However, even in the NIR region, the light depth penetration is limited, which may lead to the partial ablation of large or deep-seated tumors. A potential solution to this is the use of high-power light sources, but this may damage neighboring healthy tissues.

Thereby, integrating heating capabilities, namely MHT and PTT, into a unique nanosystem, presents several advantages. When MNPs are exposed to an alternating magnetic field, local heat is induced by magnetic energy losses [64,65]. This phenomenon, called magnetic hyperthermia, does not present the depth penetration limitations that light has, as biological tissues are transparent to magnetic fields and the use of GbMNPs allows the synergistic improvement of hyperthermia properties of MNPs, due to high thermal conductivity of GO [66,67]. Thus, GbMNPs can be accumulated into the malignant tissues to generate localized heat when subjected to NIR light or/and magnetic fields, thus preventing damage to healthy cells [67].

The therapeutic modality most often described in the studies that were reviewed is chemotherapy. Unlike standard cancer treatments that are based on chemotherapeutics and present serious adverse side-effects on healthy tissues due to the non-specificity of the drugs, carefully designed NPs can provide higher treatment efficacies and reduced side-effects through an increase in the specificity of the anticancer pharmaceutical agents towards tumors [68]. Furthermore, NPs can also enhance the pharmacokinetic and pharmacodynamic properties of encapsulated drugs compared with free drugs [68]. Once graphene-based materials exhibit delocalized *π* electrons, as well as polar chemical groups and negative surface charge, this grants high drug loading ratios for a variety of molecules [17]. In the reviewed studies the encapsulation efficiency could reach 95.8% for GbMNPs [20], which is exceptionally high compared to any other drug nanocarrier system described in the literature [69]. Doxorubicin (DOX) is the most studied chemotherapeutic drug [20,39,40,41,42,70]. Nevertheless, besides a high drug loading, the specific and timely delivering of the drugs to the targeted cells or tissues also have to be taken into account to achieve a successful therapeutic effect [4]. There are different possible strategies to achieve this goal. The most popular strategy in the reviewed studies is a pH stimuli-responsive controlled drug release, triggered by the acidic pH values found around 6.5 in the tumor microenvironment (Figure 5a) [11,20,40]. Mostly this phenomenon is attributed to the π-π stacking between the graphene-based nanostructures and the aromatic drug molecules, which can be easily disrupted under a mild acidic environment and/or the increased solubility of medicines caused by the protonation process. It is interesting to note that in some of these studies the combination of magnetic hyperthermia and chemotherapy has been shown to have a synergistic effect caused by the high thermal conductivity of graphene-based materials on cancer, when compared to the therapeutic effect when these treatment procedures are applied alone (Figure 5b) [11,20]. Another strategy used in the reviewed studies is magnetic targeting. In a study by Shirvalilou et al., IUdR/NGO/PLGA magnetic nanoparticles are developed and used with the help of magnetic targeting to treat C6 glioma. The results show a high-level of specificity to C6 glioma achieving high nanocomposite accumulation at the targeted tumor site using a magnet [43].

### 2.3. Biocompatibility and Toxicity of Graphene-Based Magnetic Nanoparticles

From the studies reviewed it is evident that the graphene-based nanomaterials’ shape, layer number, size, purity, surface properties, dose, composition and chemistry, including hydrophilicity, all play a crucial role in determining their interactions with cell membranes, cellular uptake and fate [17]. For example, lateral dimensions can promote cellular uptake (<100 nm), while larger GbMNPs are detected by the immune system, removed from the blood and discarded from the body by the liver and the spleen [46,71]. However, it is common that some of this information about each nanomaterial’s size, thickness, surface charge or even the colloidal stability, is missing in some of the published papers [40]. Therefore, a complete collection and analysis of the data regarding the graphene-based materials and their biocompatibility and toxicity to cells and tissues is difficult to archive. Nevertheless, there is a general consensus in the scientific community that among the graphene-based materials, high negatively charge rGO and GO are the ones with the best biocompatibility to cells and tissues, mostly due to their oxygenated functional groups, which allows electrostatic interactions between these graphene-based nanomaterials and the lipids of the cell’s membranes [72]. On the other hand, it has been shown that pure graphene material, which have no charge on the basal plane, do not interact with phospholipids of the cell’s membranes, but have hydrophobic interaction with their lipids, which can lead to membrane disruption [72]. Overall, these findings pinpoint the importance of the chemical surface of the graphene-based nanomaterials, which have to be taken into account for biomedical applications. Therefore, an important requirement for biomedical applications is related to the surface functionalization of the GbMNPs, which besides having the potential to improve the biocompatibility of graphene-based nanoparticles, can also improve their colloidal stability.

In biological media, it has been shown that nonfunctionalized GbMNPs tend to form aggregates due to strong intermolecular interactions, which result in unwanted changes to their physicochemical properties. Functionalization is then used as a strategy to improve the colloidal stability of GbMNPs, assuring a better body distribution and reduction in toxicity. Due to their large surface area and chemical composition, GbMNPs can be functionalized either covalently through hydroxide, epoxy or carboxylic acid groups or noncovalently through the surface π electrons, hydrophobic interactions or electrostatic interactions [18]. The functionalization of GbMNPs includes an extensive variety of strategies, namely, stealth strategies where GbMNPs are coated with polymers that inhibit protein binding and provide long-term chemical stability. However, also, using targeting strategies, where GbMNPs can be conjugated with ligands and/or antibodies that recognize specific cell surface receptors, or other strategies, where GbMNPs can be conjugated with drug and genes to their specific delivery into the target-cells. These post synthesis procedures allow GbMNPs to gain optimal characteristics for theranostic applications.

In the reviewed articles, the most common surface functionalization is PEG [20,40,45,47,49]. However, some other strategies are boarded in those articles, including coating the GbMNPs with chitosan [41] and poly (lactic-co-glycolic acid) (PLGA) [43,49]. In those works, PLGA was selected due to its potential utility in reducing enzymatic degradation, the sustained release of various drugs, the enhancement of the drug payload and the reduction in immune reactions [49]. Chitosan, i.e., polysaccharide of natural origin, was in turn selected because it responds to microenvironmental pH-stimuli by changing its solubility accordingly to the pH values of healthy and cancer cells. Additionally, it has recognized properties such as biocompatibility, biodegradability and antibacterial activity [41].

In regard to the in vitro toxicity evaluation, most of the studies reviewed use MTT [3-(4,5-dimethylthiazol-2-yl)-2,5-diphenyltetrazolium bromide] assay for cell viability evaluation in 2D static cultured cells [41,42,43]. However, this colorimetric assay is prone to producing false-positive/false-negative results, since any remaining GbMNPs in the solution to be analyzed can interfere with cell viability and proliferation results, due to the absorbance of light by the graphene-based shell [73]. Other factors to take into account when performing in vitro tests are the selection of the cell lines, the cell densities and media temperature, that could affect the nanoparticle cellular uptake [74]. These can explain the biased results observed from in vitro and in vivo studies. For instance, in some of the revised articles, although the in vitro studies showed good cell viability and high efficiency as a theranostic nanosystem, the in vivo studies revealed lower efficiency with the bioaccumulation of GbMNPs in the liver and spleen [46,71]. Additionally, it is important to empathize that the different synthetic procedures used for the preparation of these graphene-based nanocomposites use, in general, several chemical oxidizers and reducing agents, which typically generate metallic impurities and contaminations that can result in cell damage or apoptosis [72]. To overcome this synthetic limitation, green synthesis routes can be an interesting approach to reduce possible cytotoxicity of GbMNPs designed for biomedical applications.

### 2.4. Limitations of Graphene-Based Magnetic Nanoparticles for Potencial Clinical Applications

In a similar way to many others nanotheranostic systems, the lack of translation of GbMNPs to clinical applications can be ascribed to two main types of challenges, namely (i) biological and (ii) commercial ones [11].

Among the biological challenges, the evaluation of the biocompatibility and toxicity of nanostructures can be demanding, since a standard research protocol to evaluate the toxicity of a nanomedicine system is lacking, which creates bias in the comparison and standardization of biological results, limiting the development and applications of nanoparticles [75]. In the case of GbMNPs, this can be even more challenging, since there is a lack of consensus in the categorization of the many types of graphene-based nanomaterials (i.e., graphene, graphene oxide, reduced graphene oxide or even graphite). In the literature, although works performing biological studies on graphene-based nanomaterials can be easily found, not all of these studies present enough physicochemical characterization of the nanomaterials to provide clear information on the type of GbMNP developed. This is particularly important to assess, since the biological properties of GbMNPs can be very distinct if the carbon-based material used is graphene, graphene oxide, reduced graphene oxide or graphite. Additionally, the shape, size, configuration and surface charge of the GbMNPs are of the utmost importance for the biological evaluation. However, many of the articles present gaps in the data regarding size and thickness, as well as surface charge. Nevertheless, it is important to note that some graphene-based materials have demonstrated a certain dose-dependent toxicity to cells that must be considered for biomedical applications [76]. To improve reproducibility, increase quantitative comparisons of bio-nanomaterials and facilitate meta analyzes, some authors suggest the adoption of a minimum standard of information [77].

Among the commercialization challenges, one of the main problems is the difficulty of formulating a controllable and reproducible product [11]. Ideally, for large scale production, GbMNPs have to be developed in a simple and reproduceable way, with low physicochemical variability. For pharmacological interest, laborious and expensive procedures are rarely worthy of interest.

Therefore, and despite the incredible breakthroughs that nanoscience and nanotechnology have attained in the last decade, further research has to be devoted to achieving these goals, including: (i) the standardization of the production of the different graphene-based materials; (ii) the development of reproducible GbMNPs preparations; (iii) the standardization of characterization and functionalization protocols; (iv) the understanding of the nano-bio interface; (v) the understanding of the nanoparticles’ biodistribution; and (vi) the evaluation of toxicity in a wider context, in order to obtain robust and precise toxicological data [73,78].

## 3. Future Perspective for the Advance of GbMNPs

As abovementioned, despite the evident benefits and potential of GbMNP nanosystems, there are still several challenges that need to be surpassed before a clinical translation is possible. Without these advancements, graphene-based nanomaterials will face difficulties in fulfilling the rigid criteria for a significant clinical translation of the novel nanomaterials. Overall, in nanomedicine, there is a need to increase the involvement of the scientific community with the regulatory authorities. Since the commercialization of nanomedicines is mainly monitored by government policies, clear regulation and guidelines defined by the stakeholders and regulatory agencies can have a dramatic impact in the translation of nanotheranostic systems, such as GbMNPs. Another major achievement can be obtained with the effort to understand, in the early stages of material development and preclinical studies, the impact of nanomedicines in the human body. Until now, the majority of preclinical tests have been performed in static 2D cell cultures and small animals, such as mice, rabbits, or other animal species. Although they have all demonstrated their validity in the past, none of these models truly represent the patient biology and the disease heterogeneity found in humans. This subject has been mentioned as one of the major challenges in nanomedicines translation to clinics. To overcome this drawback, in recent years, advanced microfluidic devices comprising organ models, i.e., organ-on-a-chip, are being proposed. These advanced microfluidic platforms have shown high potential to reduce the bias between the results derived from preclinical and clinical trials. Furthermore, these advanced microfluidic tools are being highlighted as ideal platforms to predict the performance and biotoxicity of the developed nanomedicines, with several advantages over the gold standard methodologies [78]. Thus, organ-on-a-chip can impact the advancement of GbMNPs by providing clues in new formulations and their performance regarding particle–tissue interaction, toxicity, cell uptake, accumulation, cell targeting, transport, among others. More details on this theme can be obtained in a review found elsewhere [78].

Overall, GbMNPs are a promising nanosystem with great potential for theranostic applications. However, as debated in this review, the advancement of GbMNPs for clinical applications is intrinsically associated with other developments in nanomedicine and nanotheranostics. Nevertheless, a great effort is being made by the scientific community and stakeholders in this regard. Thus, advances in nanosystems in combination with mathematical approaches such as machine/deep learning can accelerate the discovery and successful clinical translation of nanomedicines, and it is expected that in the next decade nanotheranostics will be a promising option in medicine that, together with the development of ultrasensitive and portable nanotechnology-based devices, can revolutionize personalized care, tracking disease profiles in early stages and monitoring post-treatment recurrences, in order to predict the patient’s response to therapies and evaluate the prognosis at low cost.

## Figures and Tables

**Figure 1 nanomaterials-11-01073-f001:**
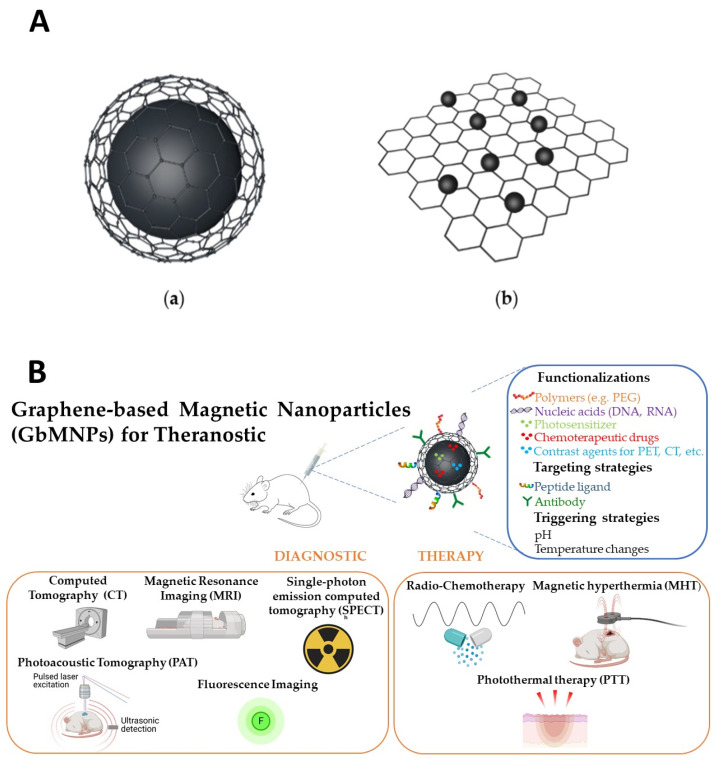
(**A**): Illustration of two basic configurations of GbMNPs. (**a**) Graphene-based materials encapsulated magnetic nanoparticles (GbEMNPs) and (**b**) graphene-based materials decorated with magnetic nanoparticles (GbDMNPs). (**B**): Schematic representation of graphene-based magnetic nanoparticles (GbMNPs), their usual functionalization, targeting and triggering strategies, as well as their most used combined diagnostic and therapeutic applications (theranostic) described in the literature. Created with BioRen-der.com.

**Figure 2 nanomaterials-11-01073-f002:**
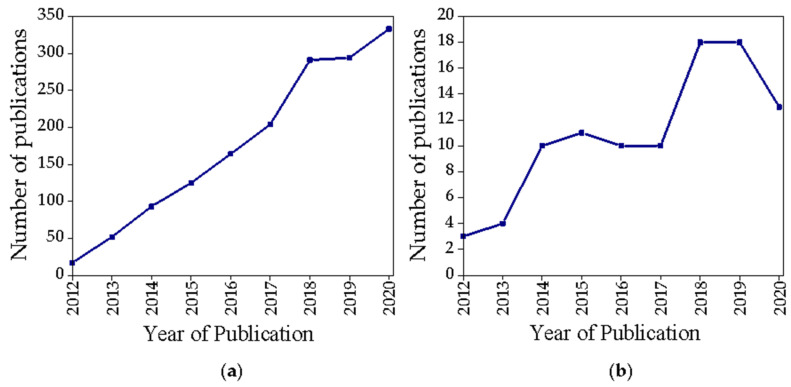
Scopus publication search results for the terms (**a**) “graphene + magnetic + theranostic” (all fields), (**b**) “graphene + magnetic + theranostic” (title/abstract/keywords).

**Figure 3 nanomaterials-11-01073-f003:**
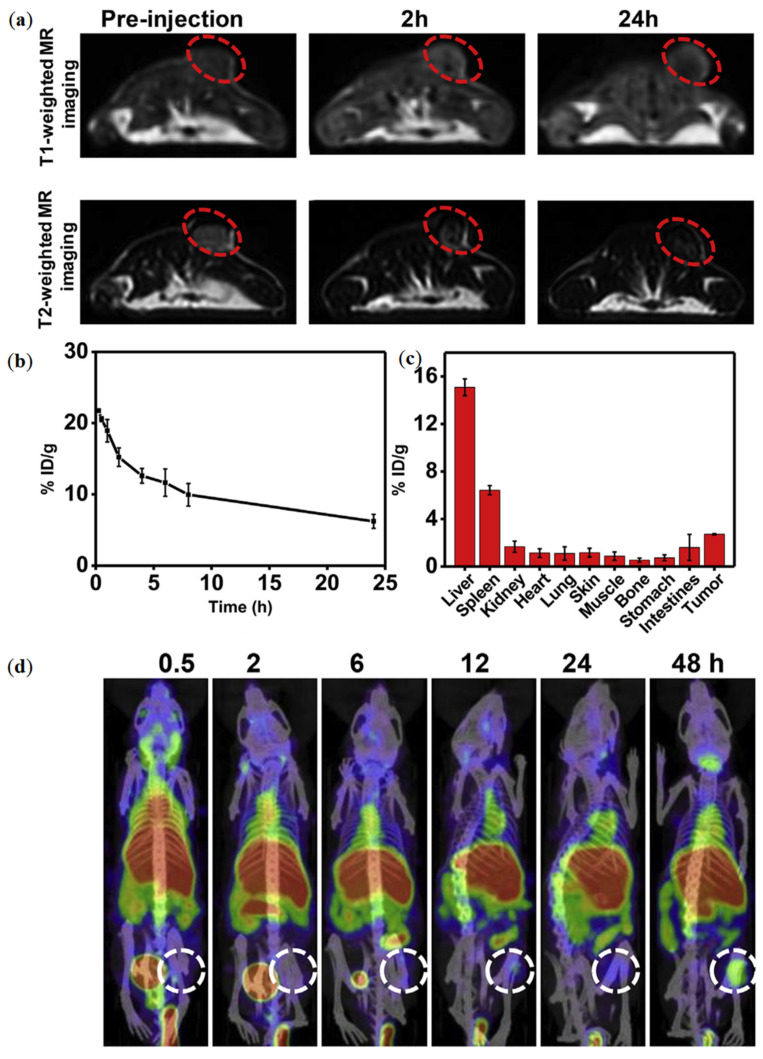
(**a**) MR imaging of mice treated with RGO-MnFe_2_O_4_-PEG nanocomposites; (**b**,**c**) in vivo performance of RGO-MnFe_2_O_4_-PEG nanocomposites. (**b**) The blood circulation and, (**c**) biodistribution of ^125^I-RGO-MnFe_2_O_4_-PEG nanocomposites. (**d**) SPECT/CT imaging of mice treated with ^125^I-RGO-MnFe_2_O_4_-PEG nanocomposites. Reprinted with permission from ref. [39]. Copyright 2019 Elsevier.

**Figure 4 nanomaterials-11-01073-f004:**
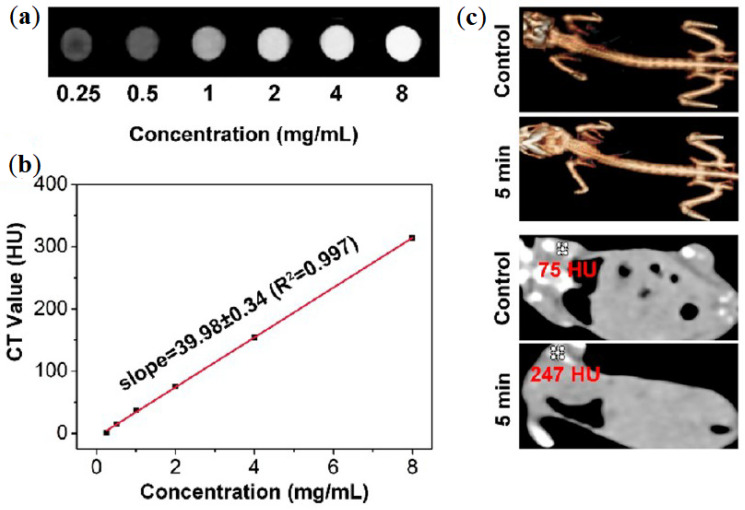
(**a**) In vitro results from CT using GZUC-PEG at several concentrations. (**b**) CT value of GZUC-PEG dispersed in aqueous solutions at several concentrations. (**c**) CT imaging and 3D renderings of CT images of tumor-bearing mice without, i.e., control, and with GZUC-PEG. Reprinted with permission from ref. [45]. Copyright 2018 American Chemical Society.

**Figure 5 nanomaterials-11-01073-f005:**
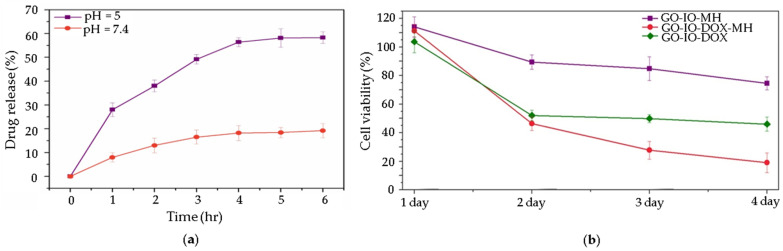
(**a**) pH dependent release of DOX from GO-IO-DOX over time; (**b**) effect of repeated periodic hyperthermia (15 min/24 h) on the CT26 cell line, used as cancer model. Reprinted from ref. [20].

**Table 1 nanomaterials-11-01073-t001:** Summary of characteristics of GbMNPs for theranostic applications, including configuration, preparation method, Dh (Diameter by DLS), Dc (Diameter by TEM), relaxivity, SAR (specific absorption rate), drug loading capacity, pH sensitive release, diagnostic strategy, therapeutic strategy, in vitro and in vivo trials.

Graphene-Based Magnetic Nanoparticle	Configuration	Preparation Method	Dh	Dc	Relaxivity (mM^−1^S^−1^)	Magnetic Properties	Drug Loading Capacity	pH Sensitive Release	Diagnostic Strategy	Therapeutic Strategy	In Vitro	In Vivo	Ref.
rGO-MnFe_2_O_4_-PEG	GbDMNPs	In situ: hydrothermal reaction	-	-	*r*_1_= 11.74; *r*_2_ = 295.48 (3.0 T)	-	-		MRI; SPECT	RIT; Chemotherapy	x	x	[39]
CoFe_2_O_4_/GO	GbDMNPs	In situ: Sonochemical reaction	-	L = 300–2000 nm; D = 5–13 nm	*r*_2_ = 92.71 (3.0 T)	Ms = 38.7 emu/g	=1.08 mg/mg		MRI	Chemotherapy	x	-	[40]
GO-IO-DOX	GbDMNPs	Ex situ: hydrophobic–hydrophobic	-	L = n/d ; D = 20 nm	*r*_2_ = 84.0 (3.0 T)	Ms = 25.37 emu/g	-	pH 7.4 = 19%; pH 5 = 58% (6 h)	MRI	MFH; Chemotherapy	x	-	[20]
MNP/GO/chitosan	GbDMNPs	Ex situ	-	-	*r*_1_ = 6.99; *r*_2_ = 44.51 (1.5 T)				MRI	Chemotherapy	x	-	[41]
GO-CD/Fe@C	GbDMNPs	Ex situ	267–334 nm	-	*r*_2_ = 9.37 (3.0 T)	Ms = 2.45 emu/g	=3.50 mg/mg	pH 7 = 20%; pH 5 = 38% (48 h)	MRI	Chemotherapy	x	-	[42]
IUdR/NGO/SPION/PLGA	GbEMNPs	Ex situ: emulsion solvent evaporation	71.8 nm	D = 26.3 nm	-	Ms = 15.98 emu/g	-	pH 7.4 = 67% (8 h)	MRI	Chemotherapy; PTT	x	x	[43]
GO/ZnFe_2_O_4_/UCNPs	GbDMNPs	Ex situ: electrostatic interactions	400 nm	L = n/d ; D = 12 nm; T = 1- 2 nm	*r*_2_ = 24.84 (1.2 T)	-	-	-	UCL; CT; MRI; PAT	PDT	x	x	[44]
GQDs-Fe/Bi	GbEMNPs	Ex situ	>100 nm	D = 64± 5.46 nm	*r*_1_ = 2.37; *r*_2_ = 62.34 (1.5 T)	Ms = 48.59 emu/g	-	-	CT; MRI	PTT	x	-	[45]
CAD-SPIONs@GO	GbDMNPs	In situ	-	L= n/d; D = 5 nm	*r*_1_ = 3.17 *r*_2_ *=* 8.36 (0.5 T)	Ms = 10.50 emu/g	-	pH 7.4 = 13.7% (48 h); pH 5.5 = 35.4% (1 h)	MRI	Chemotherapy	x	x	[46]
GO-Fe_3_O_4_	GbDMNPs	Ex situ	76 nm	L = 265 nm	*r*_1_ = 6.6 *r*_2_ *=* 71.1 (3.0 T)	-	=0.2 mg/mg	-	MRI; FI	Chemotherapy	x	-	[47]
GIPD	GbDMNPs	In situ	86.7 ± 3.4 nm	L = 150 nm ; D = 8.25 nm	-	-	=0.48 mg/mg	-	MRI	PTT	x	x	[48]
NGO-SPION-PLGA-5-Fu	GbDMNPs	Ex situ		L = 72.9	-	-	-	pH 7.4 = 41.36% (24 h)	MRI	PTT; Chemotherapy	x	x	[49]

Abbreviations: GO—graphene oxide; L × D × T—length × diameter × thickness; d—day(s); GbEMNPs—G = graphene-based materials encapsulated magnetic nanoparticles; GbDMNPs—graphene-based materials decorated with magnetic nanoparticles ; FI—fluorescence imaging; MRI—magnetic resonance imaging; CT—computed tomography; UCL—upconversion luminescence; PL—photoluminescence; PAT—photoacoustic tomography; PDT—photodynamic therapy; PTT—photothermal therapy; RIT—radioisotope therapy; Ms—saturation magnetization; n/d—not defined.

## Data Availability

Not applicable.

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
