# Peer review of "Graphene-Based Magnetic Nanoparticles for Theranostics: An Overview for Their Potential in Clinical Application"

_nanomaterials, 2021, doi:10.3390/nano11051073_

Round 1

Reviewer 1 Report

This reviewer concerns the developments of GbMNPs as theranostic probes. It could be recommended for publication after addressing the following issues:

  1. The title “Graphene-based Magnetic Nanoparticles for Theranostics: an overview for clinical application”. Actually, the clinical application of GbMNPs is still on the way. There are so many previously clinical available magnetic nanoparticles Fe3O4 withdrawn from clinical application. So, it is suggested to edit the title carefully, for example, potential clinical application, or prospect.
  2. It would be better to reorganize the contents in Figures, such as Figure 1, 2, and 6, rather than just used the same as in previously papers. Authors’ ideas or opinions or perspective are more desired here.
  3. A section containing the preparation of prepare graphene, GO, reduced GO, and GQDs as well as their GbMNPs is recommended.
  4. Section 2 “the Graphene-based Magnetic Nanoparticles for Theranostics”. Section 2.1 “the GbMNPs in diagnostic”, more contents such as PAI and USI should be included.
  5. Cytotoxicity or biocompatibility is one of major issues concerning potential clinical application. It would be better to give a discussion in this review.

Reviewer 2 Report

The authors made an interesting review about the potentialty of Graphene-based Magnetic Nanoparticles (GbMNPs) as theranostic platforms. They claim GbMNPs as materials with unique “theranostic” properties and high chemical and thermal stability. They also listed some limitations for their clinical applications.

I consider that the review is well addressed and interesting. However, there are some aspects that should be incorporated in the manuscript:

  1. As the authors said in the section 2.3., in which they showed some of the limitations of these graphene-based systems, “although works performing biological studies on graphene-based nanomaterials can be easily found, not all of these studies present enough physicochemical characterization”. And I agree… However, it is necessary to highlight what are the main conclusions of using these materiales to have a clear idea about their potentiality. I miss a section focused in the cytotoxicological aspects of using graphene-based nanomaterials to make clear, or at least have a slight idea about the influence of type of graphene-based material, size, shape and surface charge. Normally, a review of new promising materials should have a critical overview concerning this aspect since the final goal is to use them in medical applications.
  2. There are not examples of surface functionalization of these materials to increase the biocompatibility, increase the targeting properties, or induce a more therapeutic effectivity. Specific examples should be commented.
  3. I miss also more images about schematic of some of the examples. At least a representative system for imaging (diagnosis) and other for drug delivery or MHT/PTT (therapy). Or some example of a theranostic platform.
  4. Take care in Table 1 with numbers in subscript for chemical formulas (Fe2O4 especially)
  5. It would be interesting also to include a small section, or some paragraphs in the introductions for comparing the advantages of graphene-based materials in front of other carbon-based materials such as carbon nanotubes, carboranes, carbon dots…
  6. There is no mention about using GbMNPs as biosensors (for example: International Nano Letters 2018, 8, 229–239 among others), as example of diagnostic applications. The authors should be take into account this kind of applications.

In summarize, I consider this manuscript interesting. However, I consider that for being published the authors should addressed the previous comments.

Round 2

Reviewer 1 Report

This revised eidtion could be accepted for publication

Reviewer 2 Report

I consider that the authors have done an exhaustive improvement of the previous manuscript and have included different section and corrections attending the referees' suggestions made in the previous review, which has resulted in a substantial improvement in the manuscript quality. I consider that the manuscript could be accepted in the present form.